

# A low-cost acoustic permeameter
Stephen A. Drake[1], John S. Selker[2], Chad W. Higgins[2]
[1]CEOAS, Oregon State University, Corvallis, 97333, USA
[2]BEE, Oregon State University, Corvallis, 97333, USA
*Correspondence to*: Stephen A. Drake (sdrake@ceoas.oregonstate.edu)
**Abstract.** Intrinsic permeability is an important parameter that regulates air exchange through porous media such as snow.
Standard methods of measuring snow permeability are inconvenient to perform outdoors, fraught with sampling errors and
require specialized equipment, while bringing intact samples back to the laboratory is also challenging. To address these
issues, we designed, built, and tested a low-cost acoustic permeameter that allows computation of volume-averaged intrinsic
permeability for a homogenous medium. Permeameter elements were designed for use in snow but the measurement
methods are not snow-specific. The electronic components, consisting of a signal generator, amplifier, speaker, microphone
and oscilloscope, are inexpensive and easily obtainable. The system is suitable for outdoor use when it is not precipitating
but the electrical components require protection from the elements in inclement weather. The permeameter can be operated
with a microphone either internally mounted or buried a known depth in the medium. The calibration method depends on
choice of microphone positioning. For an externally located microphone, calibration was based on a low-frequency
approximation applied at 500 Hz that provided an estimate of both intrinsic permeability and tortuosity. The low-frequency
approximation that we used is valid up to 2 kHz but we chose 500 Hz because data reproducibility was maximized at this
frequency. For an internally mounted microphone, calibration was based on attenuation at 50 Hz and returned only intrinsic
permeability. We found that 50 Hz corresponded to a wavelength that minimized resonance frequencies in the acoustic tube
and was also within the response limitations of the microphone. We used reticulated foam of known permeability (ranging
from $2 \times 10^{-7}$ m$^2$ to $3 \times 10^{-9}$ m$^2$) and estimated tortuosity of 1.05 to validate both methods. For the externally mounted
microphone the mean normalized standard deviation was 6% for permeability and 2% for tortuosity. The mean relative error
from known measurements was 17% for permeability and 2% for tortuosity. For the internally mounted microphone the
mean normalized standard deviation for permeability was 10% and the relative error was also 10%. Permeability
determination for an externally mounted microphone is less sensitive to environmental noise than is the internally mounted
microphone and is therefore the recommended method. The approximation using the internally mounted microphone was
developed as an alternative for circumstances in which placing the microphone in the medium was not feasible.
Environmental noise degrades precision of both methods and is recognizable as increased scatter for replicate data points.





## 1 Introduction

Intrinsic permeability is the proportionality constant in Darcy's Law that describes the interconnectedness of air space in permeable media such as snow. It has long been recognized as an important parameter that regulates air exchange both within the snow pore space and between the atmosphere and the snowpack (Bader, 1939). Since permeability is difficult to measure directly, early efforts focused on developing empirical relationships that describe permeability as a function of more easily measured parameters such as snow density (Bender, 1957; Shimizu, 1970; Martinelli, 1971). These convenient formulations may be accurate for the specific snow conditions used to establish the empirical formula but often fail for other snow conditions (Domine et al., 2013). In an effort to obtain more accurate field measurements of permeability several experimental efforts employed vacuum flow-through devices (Conway & Abrahamson, 1984; Chacho & Johnson, 1987; Hardy & Albert, 1993; Albert et al., 2000; and Courville et al., 2007). However, flow-through measurements are time consuming and fraught with potential sampling errors and alteration of the sample from its native condition (Sommerfeld & Rocchio, 1993). Some sampling issues have been resolved by sample extraction and examination in a lab environment to measure pore space in terms of specific surface area by X-ray tomography and gas adsorption techniques (Kerbrat et al., 2008). But this multi-step process requires specialized equipment and still does not resolve issues of small-scale spatial heterogeneity and sample size representativeness (Albert, 2001).

In this paper, we describe the design, construction, and calibration of an *in-situ* active acoustic device that samples a measurement volume larger than that obtainable with current flow-through devices to acquire a volume-averaged estimate of intrinsic permeability. Active sampling methods of snow properties have been successfully applied in previous studies. For example, Albert et al. (2007) measured the evolution of an acoustic pulse, generated by firing a pistol blank, and compared the pulse shape with that of a simulated pulse to estimate volume-averaged permeability of an Alaskan snowpack. Kinar and Pomeroy designed (2007) and improved (2015) upon an active acoustic device that infers properties such as snow water equivalent from the differential backscatter at frequencies between 20 Hz and 10 kHz. The two methods that we present are less sophisticated than either the Albert or Kinar methods but have the advantage that no specialized equipment is required. Our design borrows elements from Ishida (1965), Buser (1986), and Moore et al. (1991).

## 2 Method

### 2.1 Design and Assembly of the Acoustic Permeameter

We assembled an acoustic permeameter from commonly available parts (Fig. 1). A Heathkit Model IG-1275 signal generator produced a sine wave of a specified frequency and amplitude that was split and directed both to channel 1 of a Tektronix TDS 1001 oscilloscope and to a 4Ω Altec A4468 speaker. Electrical components were powered from a DC-AC inverter and 12V car battery. The speaker was screw-mounted onto a ring of 3/4-in-thick medium-grade plywood. An acoustic tube was constructed using 20-cm (inside) diameter, schedule 40 PVC pipe. The acoustic tube was vertically mounted onto the



plywood ring and secured with butterfly clamps. During measurement, the tube was placed on the medium of interest with
the speaker facing downward such that sine waves emanating from the speaker would interrogate the medium. The
subsequent acoustic response was captured by a Radio Shack™ unidirectional microphone (model 3303038, frequency
range: 50 Hz-15 kHz) and directed to the second input channel on the oscilloscope. Frequency and amplitude of input and
output signals were stored on the oscilloscope USB drive. We do not have full knowledge of the characteristics of the
microphone transducer. However, the calibration is empirical and will vary with each microphone so discrepancies between
microphones are accommodated by the calibration.
**2.2 Acoustic Permeability Determination – External Microphone Method**
The first method used to measure intrinsic permeability involved using an externally placed microphone (EM), where we
placed the test media between the sound source and the microphone. The calibration setup for the EM method is similar to
Fig. 1 but with the speaker mounted in an upward orientation and the microphone mounted directly above the centerline of
the PVC tube, facing downward. Acoustic intensity at a given frequency was measured with the microphone in open air to
establish reference amplitude. Then a foam sample was placed over the end of the speaker tube and weighted at the edges to
minimize sample vibration. At each applied frequency, attenuation and phase shift were determined by comparison with the
reference measurement. We then utilized the low-frequency approximation in Moore et al. (1991, hereafter referred to as
M91), to calculate permeability. In M91, estimates of tortuosity ($\tau$) and effective flow resistivity ($\sigma_{pe}$) were iteratively
adjusted until optimal agreement between the measured and modeled value for the propagation constant ($k_b$) at a given
frequency ($f$) was obtained. Snow permeability ($k$) can then be computed from flow resistivity given knowledge or
assumptions of the snow grain shape. The theoretical basis for the M91 low-frequency approximation is described in Sect.
3.1 of this paper. An added benefit of this method relative to a flow-through permeameter is that it produces a measure of
tortuosity as well as permeability.
**2.3 Acoustic Permeability Determination – Internal Microphone Method**
A second permeability measurement method employed the microphone mounted inside the acoustic tube, facing outward, at
a fixed position (15 cm) from the open end. We refer to this as the IM method. We measured attenuation at 50 Hz relative to
an open-air value with the open end of the acoustic tube snug against the test media. At this frequency, the wavelength of the
emitted frequency is much greater than the length of the acoustic tube. Since the medium represents an acoustic barrier to the
emitted waveform, the amplitude of the transmitted waveform was progressively retarded as permeability increased.
Simultaneously, the amplitude of the reflected waveform, measured by the internally mounted microphone, increased. From
Morse (1952, hereafter referred to as M52) attenuation at a given frequency is a function of flow resistivity. Amplitude of the
reflected waveform is inversely proportional to the amplitude of the transmitted waveform so permeability was computed as
a function of the ratio between the initial amplitude and the reflected amplitude.





**2.4 Flow-through Permeability Determination**
We verified acoustic permeameter measurements with 5-cm-thick reticulated foam samples. The permeability of these foam
samples was established with a flow-through permeameter similar to Albert et al. (2000) with the exception that we utilized
three high-precision (Paroscientific 216B) absolute pressure sensors rather than two relative pressure sensors (Fig. 2). Using
absolute pressure sensors rather than relative pressure sensors required a modification of Albert's method so we briefly
describe our alternate method. For each foam type we cut a cylindrical section and spread a layer of petroleum gel around the
cut edge. We then slid the cut foam into the double-walled permeameter until the inner face of the foam was flush against the
inner wall of the permeameter as shown in Fig. 2. The gel sealed gaps between the foam and vessel wall. We cross-calibrated
the three pressure sensors before initiating airflow through the permeameter. The three pressure sensors monitored
atmospheric, inner cylinder, and outer cylinder pressures. Following Albert et al. (2000), we acquired the pressure drop
across each foam sample at 8-12 flow rates while adjusting a valve to regulate the pressure between the inner and outer
vessel to eliminate radial flow. We used the slope of the subsequent flow rate vs. pressure drop curve to derive permeability
from Darcy's law valid for Reynolds numbers less than 1:
$k = \frac{Q\,L\,\mu}{\Delta P\,A}$, (1)
where $k$ is intrinsic permeability (m$^2$), $Q$ is volumetric discharge (m$^3$-s$^{-1}$), $A$ is the sample surface area (m$^2$), $L$ is the sample
height (m), $\Delta P$ is the pressure drop (Pa) and $\mu$ is dynamic air viscosity (kg-m$^{-1}$-s$^{-1}$). Our samples were distinct in properties
from those published by Clifton et al. (2008) (Table 3.1), suggesting that intrinsic permeability of reticulated foam with the
same specified pores per linear inch (PPI) but different manufacturers (here Regicell and FXI Corporation) must be
determined experimentally. Permeability measured for the FXI samples in Table 3.1 span expected snow permeability values
ranging from lightly compacted snow (low permeability) to depth hoar (high permeability), (Arakawa et al., 2009).
**3 Theory**
In this section, we describe the theoretical basis for the two methods used to acoustically measure snow permeability.
**3.1 Theory for the EM Method**
The complex propagation constant for sound through permeable media can be written in general terms as:
$k_b = a + ib$ (2)
Coefficients of the propagation constant define the phase and magnitude of attenuation, respectively, and are defined in M91
as:
$a = \frac{\pi\,\Delta\phi}{180\,d}$ (3)
$b = \frac{Attenuation(dB)\ln(10)}{20d}$ (4)

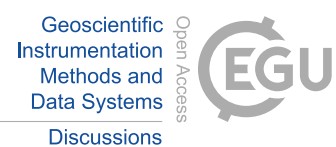

where $\Delta\phi$ is the phase shift between a free air measurement and the in-snow measurement and $d$ is media thickness. We
measure attenuation in terms of a voltage difference between the input ($V_{IN}$) and output signal ($V_{OUT}$) as:
$$Attenuation(dB) = 20\ log_{10}\left(\frac{V_{OUT}}{V_{IN}}\right) \qquad (5)$$
Attenborough (1983) developed a four-parameter acoustic model that describes acoustic attenuation through permeable
media. These four parameters: porosity, effective flow resistivity, pore shape factor and grain shape factor collapse to two
parameters in the low-frequency approximation given in M91:
$$k_b = 0.0079\sqrt{f}\left[9.10\ \tau f + i4\sigma_{pe}\right]^{0.5} \qquad (6)$$
The M91 authors used Eq. (6) to calculate the propagation constant for a given tortuosity and flow resistivity and Eqs. (2-5)
to calculate the propagation constant from phase shift and attenuation. Values for flow resistivity and tortuosity in Eq. (6)
were iteratively adjusted until the difference between the propagation constant determined by these two methods was
minimized. Due to environment-dependent nonlinear responses at both low and high frequencies the M91 authors obtained
the best fit when they used well chosen point measurements rather than measurements over a range of frequencies. Similarly,
our results are based on well-chosen point measurements. But instead of iteratively solving Eq. (6) we separate the real and
imaginary parts to independently solve for tortuosity and effective flow resistivity:
$$\tau = \frac{[a^2 - b^2]}{[9.10][0.0079]^2 f^2} \qquad (7)$$
$$\sigma_{pe} = \frac{ab}{2[0.0079]^2 f} \qquad (8)$$
We then apply empirical calibrations to match derived tortuosity and effective resistivity with known values.
**3.2 Theory for the IM Method**
With the alternate configuration of the microphone placed inside the acoustic tube, Eqs. (2-8) are no longer applicable.
Instead, we use an alternate theoretical basis to determine intrinsic permeability based on results described in M52. M52
found and Ishida (1965) verified a simple relationship between flow resistivity and signal attenuation:
$$\alpha = \beta\sqrt{\omega\sigma}\ \rightarrow\ \sigma = \frac{\alpha^2}{2\pi f}\times 10^n \qquad (9)$$
where $\alpha$ is an attenuation constant, $\beta$ is an empirical constant, $\omega$ is the imposed angular frequency and $\sigma$ is flow resistivity.
As with the EM method, we fit a sine wave to the input and output signals to eliminate white noise and then determined the
phase offset and attenuation between the input and output signals as a function of the input frequency. Instead of measuring
the attenuation through the media, we measure the amplitude of the reflected waveform at 50 Hz. We take the amplitude of
the reflected acoustic energy as an empirically-derived function of transmitted acoustic energy. Using reticulated foam
samples of known permeability, we relate the ratio of the reflected signal amplitude to initial signal amplitude with flow-
through permeability measurements.



**4 Results**
**4.1 Results for the EM Method**
Having calibrated permeability of the reticulated foam samples with the flow-through permeameter (in Fig. 2) we
subsequently obtained acoustic measurements by the EM method. Three replicates were acquired for each foam sample at
each frequency. After some experimentation, we opted to utilize data acquired at 500 Hz because this frequency lies within
the valid range of frequencies for the low-frequency approximation and because the 500 Hz results were more replicable
than those acquired at higher frequencies.
In M91 flow resistivity and tortuosity were determined by iteratively finding the best fit between measurements given in
Eqs. (2-5) and the theoretical prediction given in Eq. (6). Since we know *a priori* the foam tortuosity and permeability we
compared the known effective flow resistivity with the effective flow resistivity computed from Eqs. (2-8). As in Albert
(2007) and using a pore shape factor ratio ($s_f$) of 1, the effective flow resistivity as derived from permeability is:
$$\sigma_{pe} s_f^2 = \frac{\mu}{k} \qquad\qquad (10)$$
For the foam samples, we found that a linear fit resolved the difference between computed and known flow resistivity for a
given permeability. In short, acoustically derived effective flow resistivity did not match the actual effective flow resistivity
but deviated by a linear function from it. Therefore, from Eq. (10), we computed flow resistivity is a linear function of
permeability.
In Fig. 3 we plotted the results of three independent data sets, each data set corresponding to a different volume setting on
the signal generator and rendered in a different color. For a given data set, the signal generator volume was established for
the first data point and not changed for the remaining data points. Data corresponding to the blue/black/red data sets had
initial imposed volume that associated to RMS signal input voltages of 1.3/2.3/5.6V, respectively. Plotted in log/log format,
data points for each data set in Fig. 3 are roughly collinear and the slope of the best-fit line for a given data set is nearly
parallel to the other two data sets but the data sets are not coincident (Fig. 4). This result indicates that as the amplitude of
the imposed sine wave was increased, absolute attenuation of the signal also increased. Since the acoustically derived
permeability is sensitive to the amplitude of the imposed waveform, it is therefore important to use the same amplitude
waveform in the calibration as in subsequent permeability measurements. The relationship between $\sigma_{pe}$ and $k$ reversed at the
highest permeability in Fig. 3. The reason for this reversal is possibly attributable to non-linearity in microphone response
and represents an upper limit to the permeability that can be measured by this technique within the constraints of our
experimental design. Triplicate data points were nearly coincident in most cases; however, the blue and black asterisks in
Fig. 3 correspond to points that were more than a standard deviation removed from the other two points in the given
triplicate measurement. Differences in the imposed amplitude for these two data points does not account for their deviation
from the mean suggesting the source of error was environmental. These data were acquired outdoors in a relatively quiet but
acoustically uncontrolled location so we attribute this error is to intermittent external noise. It is therefore also important to
acquire multiple data points for each measurement so that one can distinguish the impact of incidental, external noise.



Uncalibrated tortuosity calculated for different foam samples was nearly very similar but unrealistically high with an average
of 17.7 and standard deviation of 1.54 between all of the samples. A slight but monotonic decrease in measured tortuosity
with decreasing permeability accounted for most of the standard deviation as opposed to random scatter. For comparison,
analogous data from Alvarez-Arenas et al. (2006), Kino et al. (2012) and references in Melon and Castagnede (1995) and
Doutres and Atalla (2012) note that reticulated foam commonly has a narrow range of tortuosity from 1.03 to 1.06.
Tortuosity of the FXI reticulated foam is not independently measured but shares other specifications with these other sources
and likely has tortuosity in the same range.
We empirically compensate for the differences between the theoretical and experimentally derived values for tortuosity and
effective flow resistivity. Our rationale is that the functional relationships between the theoretically and derived values for
tortuosity and effective flow resistivity are the same so the differences in values can be resolved by a first-order calibration.
Calculated tortuosity depended very weakly on media permeability so multiplying the calculated tortuosity by an
empirically-derived constant renders the theoretical value. On the other hand, calculated permeability varies linearly (on a
log scale) with effective flow resistivity (as shown in Fig. 4) so a linear calibration is required to calculate permeability from
measured effective flow resistivity. The multiplicative constant for tortuosity is found by dividing the mean known tortuosity
by the mean derived tortuosity. The linear calibration relating effective flow resistivity to permeability is taken from the
slope of the line in Fig. 4. In practice, one must calibrate the acoustic permeameter with media of known permeability and
tortuosity to derive these multiplicative constants. Once these multiplicative constants have been determined for the acoustic
permeameter, the tortuosity and permeability of other media having similar impedance can be derived without *a priori*
knowledge of media tortuosity or permeability. For the 5.6V data set we find the multiplicative constant as: $\alpha = 0.057$. For
the same data set a linear fit between measured flow resistivity and known permeability is given by:
$$log_{10}(k) = -2.0219 \times log_{10}(\sigma_{pe}) - 0.1764 \tag{11}$$
With these modifications, the M91 method obtains derived tortuosity ranging from 1.02 to 1.07 for the foam samples with a
standard deviation of ±0.03. Percentage normalized standard deviation for permeability measurements ranged from 2.1% for
the Z80 foam to 11.6% for the Z30 foam with a mean of 6.2% across all foam samples. We normalize the standard deviation
because this error measure does not change when effective flow resistivity is calibrated with a multiplicative constant. The
average percentage error between known and derived permeability was 17% by the EM method.
**4.2 Results for the IM method**
Applying Eq. (9) to results obtained with the microphone placed inside the acoustic tube at 50 Hz, 100 Hz, and 250 Hz
yielded results shown in Fig. 5. Since we are measuring reflected acoustic energy rather than transmitted energy the result
shown in Fig. 5 is an indirect (relative) measure of flow resistivity. Triplicates were acquired for each foam sample and these
data were highly reproducible as shown by the close grouping of each triplicate. Data at these three frequencies were
acquired on different days, each with a slightly different amplitude baseline (free air) setting, accounting for the horizontal
displacement between curve fits. The percentage normalized standard deviation for 50 Hz measurements ranged from 0.45%




for the Z50 sample to 19% for the highest permeability (Z10) sample with a mean of 10% across all foam samples. The
average percentage error between known and derived permeability was 10% by the IM method. Since data at 50 Hz show
improved precision relative to 100 Hz and 250 Hz data, as given by the wider domain of reflected signal amplitude, we use
data at 50 Hz to compute permeability. The piecewise spline fit (black, in Fig. 5) describes the straight-line fit between
acoustically derived resistivity and the red dashed line delineates a smooth curve used to calibrate measurements to flow-
through permeability. There is a measurable phase shift with decreasing permeability of the test media, however, M52 does
not offer a relationship for tortuosity so none was attempted with this method. Microphone response limitations precluded
measurements below 50 Hz.
**4 Discussion**
Derived permeability using the EM and IM methods are compared with the flow-through (known) permeability in Fig. 6.
Intrinsic permeability for each foam sample is given in Table 3.1. Ideally, all measurements would fall on the diagonal line
in Fig. 6 meaning that acoustically measured permeability equals known permeability for each foam sample. We find
satisfactory agreement with known permeability for both acoustic methods. The average percentage error for the EM method
was 17% and 10% for the IM method. However, the curve fit for the IM method is nonlinear so a small error in effective
flow resistivity for a high permeability medium has a greater impact than the same error for a low permeability medium.
The EM and IM methods for determining permeability each have advantages and disadvantages. One advantage of the EM
method is that the function relating effective flow resistivity to permeability is linear, unlike the more complicated
relationship for the IM method. An advantage of a linear function is that a small error in measuring the x-coordinate yields
an error that is invariant with relative magnitude when determining the y-coordinate, regardless of the magnitude of the
measurement. This behavior is not true for non-linear functions such as the more complicated function employed in the IM
method. A disadvantage of the EM method is that it is not valid for high permeability media over the range of parameters
that we tested (e.g. 5 cm media thickness). The IM method suffers decreasing precision at high permeability because the
slope of the curve is increasing but, unlike the EM method, the curve fit remains valid. Data acquired using the IM method
was highly reproducible for all of our samples, which to some extent compensates for the slope-induced decrease in
precision at high permeability. Another potential disadvantage of the EM method is that the microphone must be placed in
the media to be measured. Depending on the circumstance, emplacement in the media may be difficult or impractical.
Finally, a disadvantage of the IM method is that it is more empirical than the EM method, which may limit its applicability
to different media types. Due to the linear functional form of the EM method, we suggest that it is the preferred method as
long as the media to be measured has a permeability that is within the valid range of measurements given by the calibration.
The results presented in this paper apply directly to reticulated foam, which has similar microphysical characteristics as snow
(Schneebeli and Sokratov, 2004; Clifton et al., 2008). Nevertheless, foam is not snow and we have yet to establish how
differences in characteristic impedance between reticulated foam and snow will affect signal attenuation measurements. The
next step is to validate acoustically derived measurements in snow by comparison with other methods. Given that


characteristic impedance of snow varies linearly with snow density (Marco et al., 1998) we anticipate that the calibration
curves for both the EM and IM methods will have similar shape to the curves in Figs. 4 and 5 but with a modified slope for
snow.

## 5 Conclusions

We conclude that it is feasible to derive intrinsic permeability with a low-cost acoustic permeameter by either the EM
method or the IM method. This system returns an unambiguous, volume-averaged permeability consistent with snow
densities ranging from lightly compacted snow to depth hoar. The EM method returned intrinsic permeability within 17% of
flow-through measurements and within 2% of comparable tortuosity measurements. The IM method returned intrinsic
permeability within 10% of flow-through measurements. Measurement reproducibility by the IM method is very high such
that the standard deviation between measurements is dwarfed by environmental factors such as background noise or
inhomogeneities in the media of interest. These methods are relevant in studies for which an expedient volume average
trumps a more detailed but time-consuming vertical profile that requires measuring multiple samples using a flow-through
permeameter. One significant advantage of using these methods is that a particular frequency is measured rather than a broad
spectrum. This feature obviates the need for specialized equipment used by other methods, such as a random noise generator
and a spectral recorder/analyzer. Disadvantages of this system are that measurements are affected by environmental
conditions such as background noise and wind. A future task is to validate this system by comparison with other measures of
intrinsic permeability over a range of snow habits and densities.
**Acknowledgements**
We thank FXI Corporation for providing the reticulated foam samples that we used to perform the permeability calibrations.
Thanks also to Dr. Ziru Liu and Rebecca Hochreutener for their assistance in acquiring data.

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

3  **permeameter described in this paper.**

| Foam Type | Flow-through Permeability ($m^2$) |
|---|---|
| Regicell 10 | $160 \times 10^{-9}$ |
| Regicell 30 | $39 \times 10^{-9}$ |
| Regicell 60 | $6 \times 10^{-9}$ |
| FXI Z10 | $206 \times 10^{-9}$ |
| FXI Z20 | $137 \times 10^{-9}$ |
| FXI Z30 | $47 \times 10^{-9}$ |
| FXI Z50 | $15 \times 10^{-9}$ |
| FXI Z80 | $3 \times 10^{-9}$ |



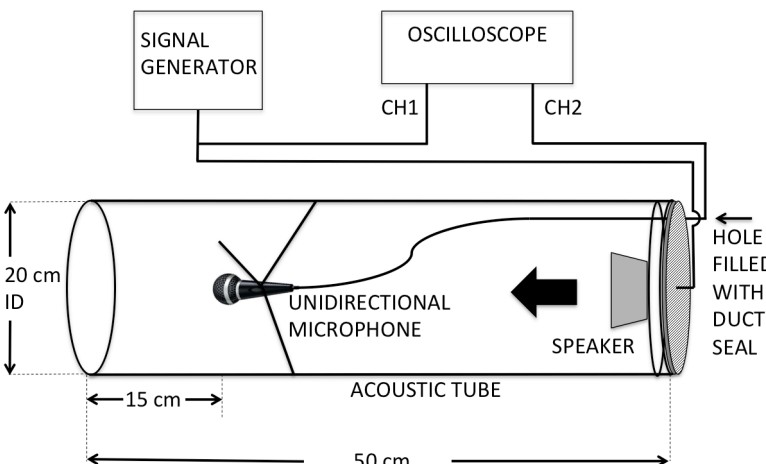

6    **Figure 1: Acoustic permeameter schematic.**


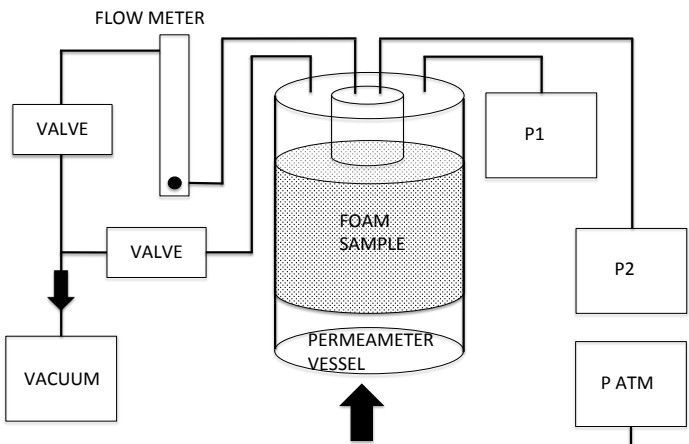

3 **Figure 2: Flow-through permeameter with three absolute pressure sensors labelled P1, P2, and P ATM.**



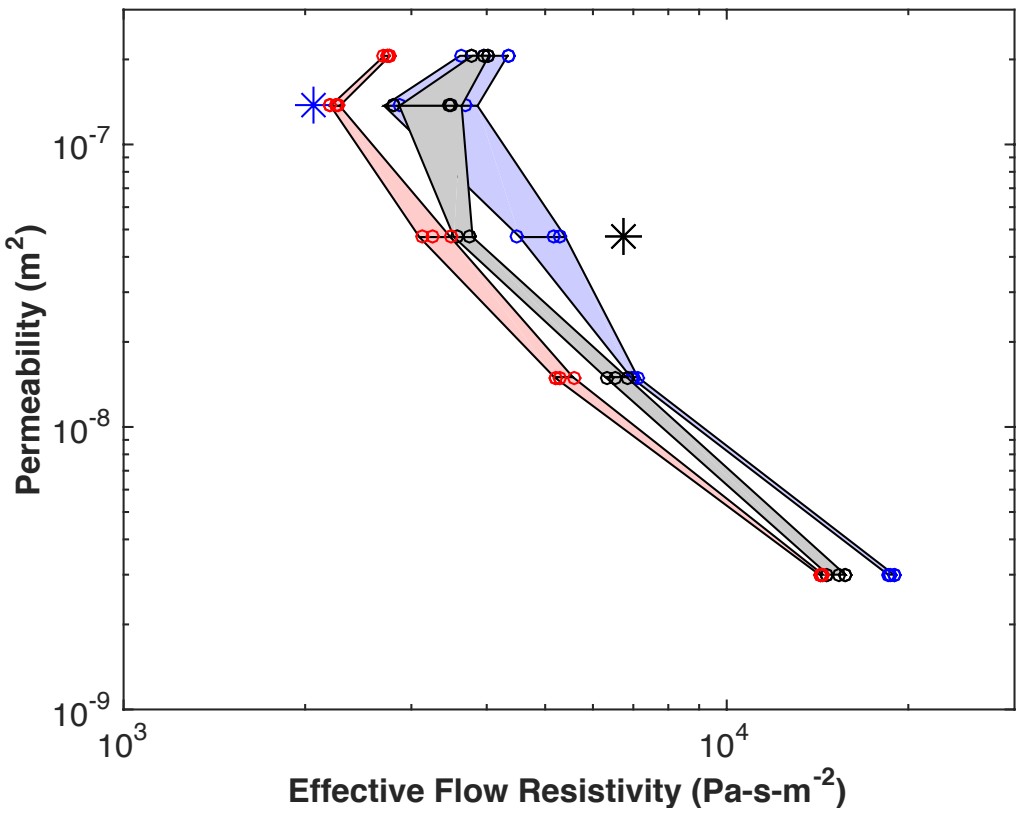

**Figure 3: Comparison of acoustically derived permeability using M91 method at 500 Hz at three different initial conditions. The**
**three initial conditions correspond to input RMS signal voltages of 1.3/2.3/5.6 volts highlighted in blue, black and red, respectively.**
**Data was acquired outdoors on clear days during periods of light winds. Color-filled regions indicate data within one standard**
**deviation of the mean. The asterisks indicate data points that exceeded one standard deviation from the mean (before they were**
**excluded) and were thus excluded from subsequent analysis.**





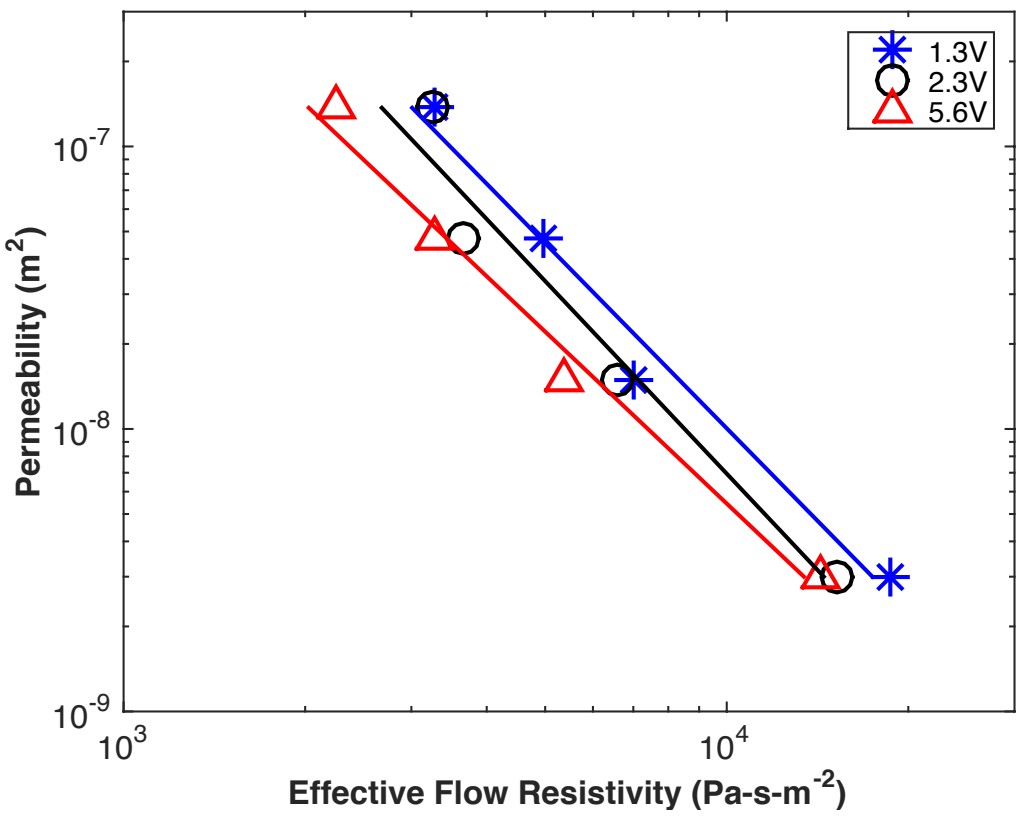

**Figure 4: Linear regressions for data in Fig. 3 with the same color for each data set. $R^2$ values were 0.98 (blue), 0.98 (black) and**
**0.99 (red).**



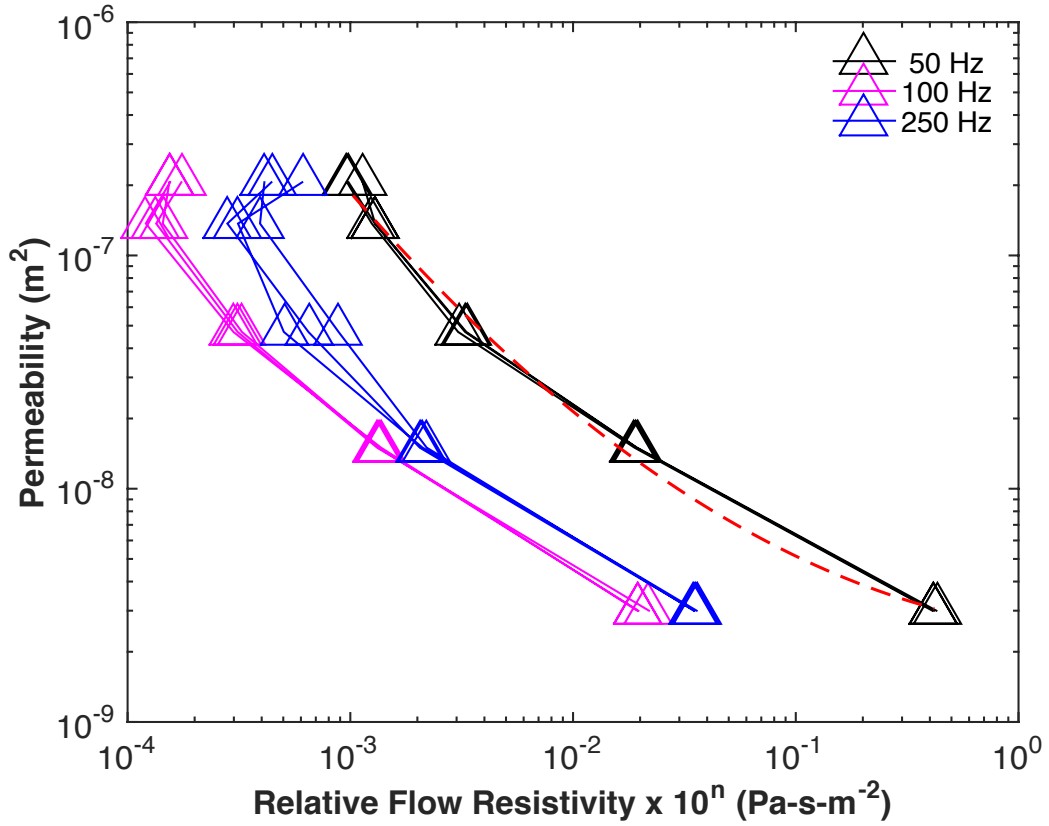

**Figure 5: Flow-through permeability as a function of relative flow resistivity using the M52 method and Eq. (9) at 50 Hz, 100 Hz**
**and 250 Hz. Three data points were acquired at each frequency and each foam type. The dashed red line defines a 2$^{nd}$ order**
**polynomial curve fit for the measurements with $p = 10^{\wedge}(0.16\,(log\,\sigma)^2 - 0.14\,(log\,\sigma) - 8.59)$.**

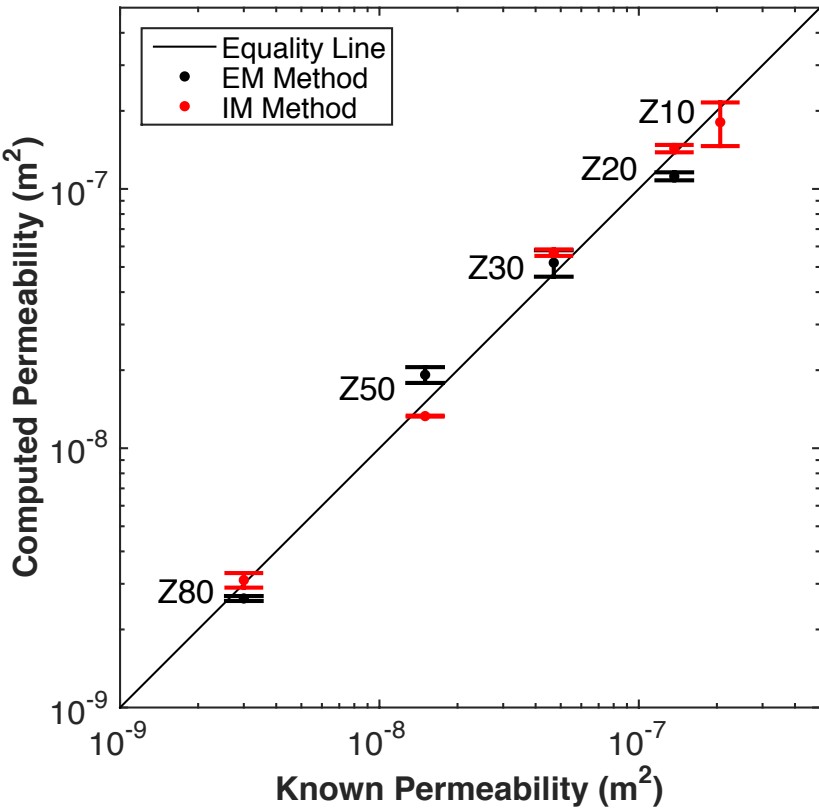

**Figure 6: Comparing flow-through (known) permeability with the EM method at 500 Hz and the IM method at 50 Hz for foam**
**samples given in Table 1. Z10 permeability was not computed by the EM method because $k/\sigma_{pe}$ is nonlinear at high permeability,**
**as shown in Fig 3. For both methods, the value for each permeability measurement is given from the curve fit for that method and**
**the standard deviation is given from the original measurements.**

