# Peer review of "A low-cost acoustic permeameter"

_Geoscientific Instrumentation, Methods and Data Systems, 2016_

## Referee Comment (RC1) · N.J. Kinar (Referee) · 14 Aug 2016

Reviewer: Nicholas J. Kinar, Global Institute for Water Security (GIWS), University of Saskatchewan, Saskatoon, Saskatchewan, Canada

A. General Comments

This paper provides techniques for measuring snow permeability using acoustics and is important since it provides an up-to-date description of how impedance tubes can be utilized for this purpose. Moreover, the techniques given in the paper provide a reference implementation that can be used as a starting point for other acoustic experiments. I highly recommend this paper to be published after some clarifications are made.

The apparatus figures (Figure 1 and Figure 2) at the end of the paper should be supplemented by photographs of actual devices in the lab and in the field. Addition of this

visual information would greatly improve the exposition of your research.

To me, it is not clear how the apparatus was deployed. Please provide a field location and time of deployment (i.e. winter or late winter). If the snowpack was layered, did the layered snowpack influence the volume-averaged measurement of permeability? Did reflections occur due to changes in acoustic impedance between snowpack layers? What was the maximum snow depth for which the device was tested? What were ambient temperature conditions at the time of measurement? Did reflections from the ground at the bottom of the snowpack influence experimental results? I am assuming that your apparatus worked well at lower ambient temperatures. Are the effects of wind on the measurement technique known? These effects are briefly stated in the caption of Figure 3, but should be discussed (i.e. a few sentences or a section) in the text. What about the effects of snow liquid water content? The presence of water in the snowpack will modify the measured permeability. Is an effective permeability being measured? I would assume that the method described in your paper works well for snow with small amounts of liquid water content.

Can you comment on impedance discontinuities related to sound waves at tube boundaries? Was the tube pressed into the snow or suspended above the surface of the snowpack? An air gap at the open end of the tube will create a discontinuity and sound pressure wave reflections should occur at this location. Are the reflections of small magnitude? If the air gap is closed by pressing the tube into the snowpack, this creates a porous media boundary condition at the end of the tube. How many tests were used to validate and calibrate the system? This is not clear from the figures in the paper and should be stated somewhere in the text.

B. Specific Comments

Page 2: Lines 2-3: To give the reader some context, you may wish to provide the equation for Darcy's Law in this section of the paper and briefly state how higher-magnitude values of permeability will increase air exchange with the snowpack. I feel

that this would be helpful for exposition, particularly for readers who are unfamiliar with the concept of applying Darcy's Law to porous media such as snow. This can be referenced later in the paper as what is now Equation 1.

Lines 16, 23, 27: The deployment of the acoustic permeameter in the field is difficult to visualize, despite the nice schematic diagram given as Figure 1. Along with Figure 1 (perhaps as Figure 1b), you could include another figure showing how the acoustic permeameter was situated to provide in-situ measurements. I find it most appealing that the permeameter can be assembled from commonly-available parts.

Line 27: The Heathkit IG-1275 is an older (analog) signal generator that probably works well. For future experiments, perhaps Direct Digital Synthesis (DDS) could be used. Can you comment here (or in another section of the paper) on oscillator drift with respect to component aging (i.e. change in capacitance values over time), temperature, jitter, and frequency accuracy? Can you measure the time-domain jitter and the oscillator phase noise and report these measurements in a quantitative fashion? Such numbers are important since they demonstrate that simple equipment can be used.

Page 3: Line 1: I am assuming that the tube was placed on the snow surface so that the end of the tube opposite from the speaker was in contact with the snow. Could you add a subfigure to Figure 1 showing how the tube was placed in relation to the snowpack? A photograph would be very informative.

Lines 3-4: Did you use phantom power for the microphone? Generally, the use of phantom power increases microphone sensitivity. Even if the microphone is not a high-end reference mic, for future experiments it can be calibrated using a pistonphone or a reference sound source. Calibration in an anechoic chamber would be useful. Can you provide an approximate sensitivity for the microphone? This is easy to provide if the voltage output of the microphone is measured with a source set to a reference frequency (i.e. 1 kHz).

Line 10: The calibration setup is difficult to visualize here. Consider adding a figure or

a picture to demonstrate.

Line 15: Rationalize here why the low-frequency approximation works well.

Lines 16-17: Which numerical scheme was used for iterative adjustment?

Line 27: Is this RMS amplitude?

Line 31: Can you clarify what is meant by "initial amplitude"? An equation might be suitable to reference rather than a description in the text.

Page 4 Line 2: Can you list the manufacturer part numbers here (and reference page 12, Table 1 of the paper)? Does the manufacturer provide reference data for foam permeability?

Page 5 Line 6: List low-frequency assumptions either here or above (see Page 3, line 15 comment).

Line 17: What are these empirical calibrations, and how were they derived? Can you provide an equation or a reference? Are these empirical calibrations necessary due to instrumentation error?

Line 24: What is the origin of the white noise? Is this due to the sampling system (i.e. oscillator and ADC) and environmental noise?

Line 26: Why is 50 Hz being used here as a reference frequency?

Line 29: It would be good to include reference to a calibration equation here showing coefficients for the experimental system.

Page 6 Line 6: What is the valid range of frequencies for the low-frequency approximation?

Line 15: Can you include the calibration equation here?

Line 17: Consider specifying the volume of the signal generator with respect to acoustic power of the source. The RMS voltage is also useful to state here, but how does this

relate to the Sound Power Level?

Line 26: What about non-linearity in the loudspeaker response? Note that loudspeakers are extremely non-linear transducers. Loudspeaker pre-emphasis techniques and equalization might be useful to use in the future.

Page 8 Lines 7-8: What is the range of snow densities, and how many snow sampling points were used?

Line 22: Is the 5 cm material thickness associated with the same accuracy and precision with respect to measurements made with actual snow?

C. Technical Comments Throughout the paper, please provide SI units for all variables. A list of nomenclature at the end of the paper would be helpful.

Page 4, Line 20: Remove comma between the closing bracket ")" and the Arakawa (2009) reference.

Figures 3, 4, 5, 6: Consider removing dashes "-" between units in the axes labels.

Figure 3: Consider providing a legend in the figure.

Figure 2: Explicitly state that "P ATM" is an atmospheric reference pressure. A reference to Hardy and Albert (1993) could be included in the figure caption.

Please also note the supplement to this comment:
http://www.geosci-instrum-method-data-syst-discuss.net/gi-2016-13/gi-2016-13-RC1-supplement.pdf

―――――――――――――――――

---

## Author Comment (AC1) · 18 Oct 2016

Thank-you Dr. Kinar for your thoughtful review.

A. Response to general comments:

The "general comment" questions of this review regard field deployment of the acoustic permeameter in snow. Although the acoustic permeameter is designed for field deployment in snow, the purpose of this paper is to explore the utility of the presented design to acoustically acquire volume-averaged permeability measurements of standard media to test the methodology. In-snow field measurements acquired with this device cannot be validated without extensive independent measurements using a currently verified method such as the flow-through permeameter referenced in Hardy and Albert (1993). We consider verification of in-snow permeability measurements by an independent device has yet more complications that are beyond the scope of this paper. For example, acoustic permeameter measurements apply to a much larger volume than flow-through permeameter measurements invoking issues with inhomogeneities such as snow layering and representative sample size that would also need to be addressed. We therefore narrowed the focus of this paper to a uniform standard media rather than snow as indicated by the title, (which does not contain the word "snow"). To underscore this point, on page 1 at line 12 we added the sentence: "In this paper, we compare acoustically-derived permeability with results derived from a standard flow-through permeameter using reticulated foam samples."

B. Response to specific comments:

Page 2, Lines 2-3: Moved equation (1) to page 2, line 5 and described Darcy's Law.

Page 2, Lines 16, 23, 27: Added Fig (2) showing a picture of the system deployed for EM calibration and renumbered subsequent figures. Added discussion relating volumetric flux and intrinsic permeability.

Page 2, Line 27: Many enhancements could be made with the parts that were used and thereby improve measurement error. We report but do not recommend particular electronic parts utilized to assemble the acoustic permeameter. One of the points of this paper is that, because the measurements are relative, frequency inaccuracy associated with drift is minimized so lower cost components can be utilized. The critical measure of the quality of the signal generator is that the imposed and modified waveforms are sufficiently robust to extract a single-valued sine wave.

Page 3, Line 1: A photograph of the setup was added as Fig. 2 that should inform this comment.

Page 3, Lines 3-4: Phantom power was used to power the microphone. Microphone sensitivity varied with frequency. Since measurements were relative the impact of frequency-dependent sensitivity was minimized.

Page 3, Line 10: Added Figure 2 showing the EM calibration configuration.

GID
Page 3, Line 15: Rationale for using the EM method is given in the "Theory" section as well as the referenced M91 paper.

Page 3, Lines 16-17: As explained in the "Theory" section, this method precludes the necessity of employing a numerical scheme (unlike the M91 methodology).

Page 3, Line 27: Added "maximum" to discriminate it from RMS amplitude.

Page 3, Line 31: Changed "initial amplitude" to "amplitude of the imposed waveform".

Page 4, Line 2: Complete model numbers for foam types were included in Table 1. The manufacturer does not give reference permeability information.

Page 5, Line 6: Changed "two parameters" to "three parameters". Added some explanation of the assumptions used by M91. A more comprehensive explanation of assumptions would require repeating the derivation in Attenborough (1983), obfuscating the narrative.

Page 5, Line 17: Added an explanation of the need for an empirical calibration. The empirical calibrations are discussed in the section titled, "Results for the EM Method".

Page 5, Line 24: replaced "white" with "both environmental and microphone-derived".

Page 5, Line 26: Appended: "a mid-point value for the low-frequency approximation".

Page 5, Line 29: This section describes the EM theory so it is not appropriate to give calibration values here. The EM calibration is given in Eq. 11 and the IM calibration is given in the Fig. (6) caption.

Page 6, Line 6: Added, "(up to 2 kHz)" as given in Moore et al. (1991) for the valid range of frequencies for the low-frequency approximation.

Page 6, Line 15: Calibration equation and coefficients are given in Eq. 11.

Page 6, Line 17: I don't understand the first part of this question. The RMS voltage is a single value measured by the oscilloscope whereas the sound power level is the ratio
of input voltage to output voltage.

Page 6, Line 26: Issues related to non-linearity of the loudspeaker response are a valid concern but were minimized by acquiring data at only 3 frequencies.

Page 8, Lines 7-8: This paper does not document snow measurements.

Page 8, Line 22: The 5-cm sample thickness was chosen because it was sufficiently thick to attenuate the measured frequencies and because it was available for all sample models from the manufacturer.

C. Response to technical comments:

Page 4, Line 20: Removed the extra comma.

Figures 3, 4, 5, 6: Regenerated figures without dashes in the x-axis label units.

Figure 3: Added a legend.

Figure 2: Described the "P ATM" sensor and added Hardy & Albert (1993) reference to figure caption.

New Fig. 2 is attached with this response. The complete caption for this figure is: EM calibration setup with the microphone suspended above a foam sample. Foam samples, the oscilloscope and the frequency generator are located on the foldup table. Not shown are a ring weight that was placed around the perimeter of the foam sample to minimize vibration and foam sheets that were placed beneath the acoustic tube to dampen amplitude of reflected acoustic energy.

---

## Referee Comment (RC2) · N.J. Kinar (Referee) · 27 Dec 2016

N.J. Kinar (Referee)

njk024@mail.usask.ca

Received and published: 27 December 2016

Reviewer: Nicholas J. Kinar, Global Institute for Water Security (GIWS)

**A. General Comments**

The paper describing a low-cost acoustic permeameter is a worthwhile contribution to the journal and should be published. However, the revision removed much of the testing related to snow, which I believe is an important part of a paper published in a journal related to geoscientific instrumentation. The measurements on snow included in the first submitted draft are interesting and should be retained since snow is a natural porous material (a geomaterial) that is not artificial and not the same as foam samples.

Since this paper is a proof-of-concept, the field site location where the snow was sampled is not of particular importance since the focus of the paper is on testing of porous materials. Similar to Figure 1, the snow could have been sampled from a backyard lo-
cation. If this was the case, it is okay to state that the snow was sampled at a backyard location, and perhaps a follow-up paper could be written testing the permeameter at a University of Oregon field site. It is important to describe how the snow was placed into the permeameter and retained for sampling. In addition, the time, date and location (i.e. city or town) associated with the snow samples should be noted to provide context. Subject to the decision of the editor, the paper can be: (a) published without reference to snow; or (b) published with some testing on snow. If the authors are willing to revise the paper to once again provide details related to testing on snow, I will quickly review the paper and perhaps this paper can be submitted for complete publication in the journal. Ideally, this could be done within the next month.

**B. Specific Comments**

None at this time, but I will add additional comments if necessary when the paper is revised to re-add the data related to snow. The authors have taken into consideration the salient points of the comments given with respect to the first draft.

Please also note the supplement to this comment: http://www.geosci-instrum-method-data-syst-discuss.net/gi-2016-13/gi-2016-13-RC2supplement.pdf

---

## Author Comment (AC2) · 25 Jan 2017

Response to general comments:

This note is a response to Dr. Kinar's comments dated 27 December 2016. Dr. Kinar requested that we include field-based measurements of snow permeability derived from the acoustic permeameter. We do acknowledge that a new set of field experiments and analysis would add another dimension to the manuscript. In its current form, the manuscript is presented as a proof of concept for generalized permeable media measurements over a permeability range consistent with seasonal snow, but snow measurement is not the focus. If readers of this comment are interested, the device used in Moore & Attenborough (1992) has parallels with our improved design, and presents measurements of snow permeability. We excluded targeted snow measurements from the manuscript to make it more general with the intention of revisiting the targeted snow permeability applications in a separate experiment. We are confident

that Dr Kinar is aware that a complete validation experiment is not a trivial endeavor. Many interdependent parameters must be categorized such as snow layering, snow microphysical characteristics, meteorological conditions etc. with an appropriate sample size, validating technique, among others.

That said, we appreciate and share Dr. Kinar's interest in improving the manuscript and concur with his assertion that a selection of field-based snow measurements would add value to the manuscript. Our conundrum is that on one hand, validated snow permeability measurements over the full parameter space of snow type and condition are beyond the scope of this paper and on the other hand additional measurements would improve the manuscript. We therefore seek guidance from Dr. Kinar as to the appropriate compromise in scope of the proposed additional measurements such that they are feasible (can be done with no financial support/travel budget) and timely (must be done before seasonal snow melts) yet add value to the current effort.

---

## Referee Comment (RC3) · N.J. Kinar (Referee) · 27 Jan 2017

I still think this is an interesting paper that demonstrates a low-cost acoustic permeameter, and I would like to see it published in this journal.

I initially believed that the permeameter had been tested on snow, and that snow data had been removed from an early draft. That is why I asked for clarification, and for snow data to be included. Some of the lines in the paper led me to believe that it had been tested on snow. A good example of this is a line in the abstract: "The permeameter can be operated with a microphone either internally mounted or buried [at] a known depth in the medium."

I think that testing the paper on some natural geomaterial would be worthwhile. Even if one or two example tests on snow would be included, that would be a good addition to the paper. A large number of snow samples with great variability may not be appropriate or within the scope of this paper, but a brief test on some natural geomaterial would be appropriate to add.

I ask the Editor to comment on this. Should the paper be published without a test on a natural geomaterial? Can a paper with only foam sample tests be included in this journal?

If the paper should be published without any further revisions, I strongly believe that the paper is interesting and the physics is reasonable.

The authors write that "Our conundrum is that on one hand, validated snow permeability measurements over the full parameter space of snow type and condition are beyond the scope of this paper and on the other hand additional measurements would improve the manuscript."

I believe that a simple test on a natural geomaterial (such as snow) would be worthwhile to add to the paper. A few sample measurements (even one sample measurement of snow permeability) would demonstrate that the permeaometer can be used for a natural geomaterial such as snow, as suggested by the authors.

---

## Referee Comment (RC4) · Anonymous Referee #2 · 15 Feb 2017

The authors had been addressing the subject properly after clarifying their objectives. As the method hold promise to be a cost effective tool for snow permeability measurements in the future it is well justified to have it published in this journal.